# Healthcare Resource Utilization, Economic Burden, and Multi-Level Medical Security System for Individuals with Spinal Muscular Atrophy in Shaanxi Province, China

**DOI:** 10.3390/healthcare13040428

**Published:** 2025-02-17

**Authors:** Mingyue Zhao, Shengjie Ding, Yuhan Zhao, Chenglong Lin, Yubei Han

**Affiliations:** 1Department of Pharmacy Administration and Clinical Pharmacy, School of Pharmacy, Xi’an Jiaotong University, 76 Yanta West Road, Xi’an 710061, China; jason_2024@stu.xjtu.edu.cn (S.D.); yh010918@stu.xjtu.edu.cn (Y.Z.); chenglong.lin@stu.xjtu.edu.cn (C.L.); yubeihan0307@stu.xjtu.edu.cn (Y.H.); 2Center for Drug Safety and Policy Research, Xi’an Jiaotong University, Xi’an 710061, China

**Keywords:** spinal muscular atrophy, healthcare resource utilization, economic burden, multi-level medical security system, China

## Abstract

**Objectives:** The objective of this study is to quantify healthcare resource utilization, economic burden, and the multi-level medical security system for Spinal Muscular Atrophy (SMA) patients in Shaanxi Province, China, from a societal perspective using a survey. **Methods**: This observational study employed an online survey with a retrospective cross-sectional design in Shaanxi Province, China. The survey examined various aspects of SMA, including resource utilization, direct and indirect economic burdens, and co-payment mechanisms within a multi-level medical security system. **Results:** Following the inclusion of nusinersen in the National Reimbursement Drug List (NRDL) in 2022, the treatment rate for SMA patients increased significantly. After risdiplam was added to the NRDL in 2023, its use also saw a marked increase. Treatment costs varied by SMA type: Type 1 incurred the highest costs (RMB 300,000 or USD 41,000), followed by Type 2 (RMB 270,000 or USD 37,000), Type 3 (RMB 200,000 or USD 27,000), and Type 4 (RMB 80,000 or USD 11,000). The primary sources of costs were productivity losses due to primary caregivers (32.94%), nusinersen usage (29.29%), and risdiplam usage (17.33%). Out-of-pocket costs for SMA patients accounted for 29.29% of the total costs. In 2023, basic medical insurance covered 49% of direct costs and 32% of total costs. Patients still had to pay 25.73% of the total cost for the direct costs. **Conclusions:** Basic medical insurance is a critical foundation for patient security and plays a pivotal role in reimbursement. In contrast, commercial insurance has a relatively limited impact on covering the costs for SMA patients. These findings highlight the substantial healthcare burden faced by SMA patients under the current healthcare system in China.

## 1. Introduction

SMA is a rare inherited neuromuscular disorder that causes progressive muscle degeneration, impacting mobility and quality of life across all disease forms. It can significantly affect life expectancy, particularly in patients with Type 1 SMA, who generally have a life expectancy of less than two years making it the most severe form of the disease [1,2,3]. Prior to 2016, treatment options were limited to symptom management [4]. However, in 2017, the Food and Drug Administration (FDA) and the European Medicines Agency (EMA) approved nusinersen, the first disease-modifying therapy (DMT) for SMA [5,6]. This was followed by the approval of onasemnogene abeparvovec in 2019 and risdiplam in 2020 [7,8,9]. These DMTs slow the progression of SMA and are most effective when administered early. Despite their benefits, the high cost of these therapies makes them inaccessible to many patients worldwide.

The affordability of drugs for rare diseases is closely tied to a country’s pricing strategies and health insurance systems. A Health Technology Assessment (HTA) is a crucial tool used globally to determine the pricing and reimbursement of innovative drugs [10,11,12]. In China, the pricing of less competitive drugs is often based on HTA evidence [13]. The drugs for treating SMA include nusinersen and risdiplam. Before 2022, although both drugs were available on the market, they were not included in the reimbursement program, and the number of patients using them was very low. When HTA evaluates orphan drugs, the assessment process differs from that of non-orphan drugs due to their unique characteristics. The evaluation considers aspects such as clinical effectiveness and social value. For example, in the case of drugs treating SMA, clinical trials of nusinersen and risdiplam have shown that they significantly improve patients’ motor function and slow disease progression. Since SMA patients are mostly children, treatment has a profound impact on their future lives, giving these drugs high social value. Real-world data further validate the long-term efficacy and safety of nusinersen and risdiplam, strengthening the evidence for their inclusion in healthcare reimbursement systems. Through national drug price negotiations, nusinersen and risdiplam were included in China’s NRDL in 2022 and 2023, respectively [14,15]. Indeed, the Chinese government has a budget in place when conducting drug price negotiations. The cost-effectiveness threshold for SMA-treating medicines is higher than other medicines based on the current policies. Furthermore, China’s latest medical security plan, outlined by the General Office of the State Council, introduces a multi-tiered medical insurance system aimed at benefiting all citizens, including those with rare diseases [16]. This system consists of three main levels: basic medical insurance, supplementary medical insurance, and social assistance programs. Basic medical insurance includes both employee and resident medical insurance, critical illness insurance, and medical assistance programs (see Appendix A) [17]. After being included in the health insurance system, most patients are now able to receive treatment. The payment for medication costs includes the government, commercial insurance, social assistance, and individuals. The commercial insurance and social assistance have a limited role in covering medication costs. In China, real-world evidence is not necessarily required for a drug to be included in the NRDL. HTA evaluations based on RCT (Randomized Clinical Trial) materials are also a method used to determine whether a drug can be included in the NRDL. For the treatment of SMA, the income and expenditure levels of medical insurance vary across different regions, leading to differences in reimbursement rates. The main sources of China’s medical insurance funds are individual contributions from employees and urban and rural residents, employer contributions, and government fiscal subsidies. These funds collectively form the foundation of the medical insurance system, which is used to meet the healthcare needs of insured individuals.

The clinical and economic burdens of SMA are substantial, affecting patients, families, caregivers, and healthcare systems [18,19,20,21,22]. These burdens are influenced by the type and severity of the disease. A systematic literature review on global direct costs found that annual expenses for Type 1 SMA ranged from EUR 67,934 to EUR 177,811 (2021 prices) [19]. Other reviews also highlighted the high direct medical costs associated with SMA [20,21,22], particularly in Europe and the U.S. before the advent of DMTs [20,22]. A study conducted in Hong Kong assessed the long-term mortality and economic impact of SMA, revealing that annual direct medical costs were highest during the first three years of life, ranging from HKD 559,503 (USD 71,731) to HKD 759,091 (USD 97,320). These costs gradually decreased with age, dropping to HKD 119,361 (USD 15,303) by the time patients reached 17 years of age [23]. Recently, a nationwide web-based study in China, focusing mainly on adult SMA patients, revealed that these patients did not receive adequate treatment during the study period [24]. However, there is limited data on healthcare resource utilization and the economic burden of SMA treated with DMTs in mainland China. This study aims to quantify healthcare resource utilization, economic burden, and the role of multi-tier insurance for SMA patients in Shaanxi Province, China, from a societal perspective through a survey.

## 2. Methods

### 2.1. Study Design

This study employed an observational, online survey with a retrospective cross-sectional design, conducted in Shaanxi Province, China. The primary aim was to investigate various aspects related to SMA, including healthcare resource utilization, the economic burden (encompassing both direct and indirect costs), and the co-payment mechanisms within the multi-tier medical security system.

### 2.2. Study Sample

SMA patients and their caregivers were recruited through the Meier Advocacy and Support Center (MASC) for SMA [25], the only professional nonprofit organization specializing in SMA in China. Participants were eligible for inclusion in this study if they were family members of SMA patients or adult patients who had knowledge of their own treatment and associated costs, demonstrated good compliance, and either resided in or received treatment in Shaanxi Province, China. Patients involved in long-term interventional clinical trials (including extension studies) were excluded from this study. If a patient was unable to read or write, their caregiver was allowed to complete the survey on their behalf. However, caregivers who could not read or write were excluded. An invitation letter, a detailed study information sheet, and consent forms were distributed to all eligible participants. We contacted MASC’s director in Shaanxi Province via email and posted a recruitment notice in WeChat groups, explaining this study’s purpose and significance. A total of 42 families were invited to participate through voluntary sign-ups and recommendations from the director. However, 5 families withdrew prior to the interviews due to concerns about the privacy of the recording process. As a result, we received 37 valid responses. To validate the sample size, we consulted the Shaanxi Provincial Medical Insurance Database, which reported 34 patients using Nusinersen in 2022, aligning closely with the 27 patients from our study that year. Among the 37 respondents, 3 were adult patients, and the remaining 34 were family members (primary caregivers). All participants provided verbal informed consent and received compensation based on the interview duration.

### 2.3. Ethical Considerations

This study was conducted in accordance with the Declaration of Helsinki and received approval from the Research Ethics Committee of The First Affiliated Hospital of Xi’an Jiaotong University (Approval No. XJTU1AF2024LSYY-218). Detailed information regarding the ethics committee approval and the review process can be found in the attached document [Appendix A].

### 2.4. Data Collection

Data were primarily collected through resource use questionnaires completed by the families of the patients, with only one adult patient participating in the questionnaire survey independently. These data are related to resource utilization for SMA patients between 2021 and 2023. The survey was conducted in October 2024. Pilot testing was conducted to ensure the clarity and appropriateness of the questions and to verify that the questionnaire captured all relevant aspects of resource use. The survey was conducted by two trained investigators who provided assistance during the questionnaire completion process, clarifying any uncertainties. After collection, the original data were reviewed and verified for completeness. Invalid questionnaires, including those filled out multiple times or with significant errors, were excluded. Valid questionnaires, defined as those with accurate and complete data, were included in the final analysis. Each interview session was audio-recorded and transcribed for further analysis.

### 2.5. Survey Instrument

The questionnaire covered five main domains: (1) demographic information, (2) clinical disease characteristics, (3) healthcare resource utilization, (4) economic burden of illness, and (5) reimbursement and insurance status. The survey was conducted using the WENJUANXING Survey platform.

### 2.6. Statistical Analysis

Healthcare costs were calculated based on actual healthcare service utilization by each patient from 2021 to 2023. The total direct medical costs were derived by multiplying unit costs by the frequency of healthcare service utilization at each patient’s age. We inquired about the annual direct costs for each patient, as well as their utilization of healthcare resources, including the number of hospitalizations, outpatient visits, and medication administrations. We calculated the total direct costs by multiplying the frequency of healthcare resource utilization by unit prices. In China, drug unit prices are publicly available, and the cost of a single hospitalization for SMA medication is approximately CNY 5000 (USD 700). Outpatient visits primarily involve registration fees for prescription refills, with minimal costs that can be neglected. The costs of hospitalizations due to complications were obtained by directly asking the patients. Productivity loss was estimated using the human capital approach, where the value was based on gross wages and payroll taxes. The primary caregivers of SMA patients are typically their parents, with mothers being the main caregivers in most cases. Grandparents may also assist in caregiving, but they are not the primary caregivers. Since SMA patients are generally young, their parents are usually not yet of retirement age. It was assumed that adult patients and caregivers would have worked full-time had the patient not been affected by SMA. Additionally, if the patient’s caregiver spent more than eight hours per day providing care at home, this was considered a loss of labor force. Family caregiving was valued using the opportunity cost approach, where the value of lost leisure time was equated to the caregiver’s net wage. The cost of individual productivity loss is calculated based on the per capita annual income in Shaanxi Province. Direct, indirect, and total costs were calculated and presented both in aggregate and by individual cost components, with corresponding resource utilization highlighted.

Descriptive statistics (mean and standard deviation [SD] for continuous variables; count and proportion for categorical variables) were used to describe the demographic and clinical characteristics of the SMA population by disease type. Due to the skewed distribution of resource use and costs, both the mean [SD] and median [IQR] values were reported. Data were entered using EpiData with double data entry and cross-checking. All analyses were performed using STATA 16.

## 3. Results

A total of 42 patient–caregiver pairs were invited to participate in this study, of which 37 ultimately took part. Among the participants, there were four patients with SMA Type 1, twenty-one with SMA Type 2, eleven with SMA Type 3, and one with SMA Type 4. The average age of the patients was 10 years. A slightly higher proportion of male patients (54.1%) compared to female patients (45.9%) was observed.

As shown in Table 1, most patients (81.2%) had purchased commercial insurance, and all were covered by China’s basic medical insurance; 75.7% also had inclusive commercial insurance. Commercial insurance typically excludes coverage for certain diseases, such as SMA, from its protection scope. In contrast, inclusive commercial insurance generally imposes no disease restrictions, features low premiums, and has low entry barriers. Although it offers broad coverage, the reimbursement rate is usually relatively low (e.g., 70–80%), and it often includes a high deductible (e.g., CNY 10,000–20,000 or USD 1400–2800). Typically, the commercial insurance does not cover diseases such as SMA. The premiums for basic medical insurance and inclusive medical insurance are relatively low. The cost of the insurance scheme was paid by the patients’ parents.

Disability affected 81.1% of the patients, with more than half (56.7%) classified as having a grade 1 disability. On average, symptom onset occurred at 2.3 years, with diagnosis occurring at 3.8 years. Scoliosis was a common complication, affecting 70.3% of patients. Families typically consulted four hospitals (the median number) before obtaining a diagnosis. Household sizes averaged four members, with 70.3% of families being married and 29.7% divorced, partly due to the challenges posed by SMA. Caregiving responsibilities were substantial, with 45.9% of families having one caregiver and 48.6% having two. The age of the primary caregiver ranged from 29 to 54 years, with an average age of 39; 69.7% of primary caregivers were female. Notably, 75.7% of primary caregivers were unemployed, dedicating an average of 19 h per day to caregiving.

In China, the subsidies or assistance available to people with disabilities vary across different regions. Typically, people with disabilities can receive living subsidies, nursing care subsidies, medical assistance, and minimum living guarantee or special hardship assistance. In Shaanxi Province, the living subsidies and nursing care subsidies are very minimal and can be considered negligible. In 2023, the rural minimum living guarantee standard in Shaanxi Province was approximately CNY 6000 (USD 840) per person per year, while the urban minimum living guarantee standard was about CNY 8000 (USD 1120) per person per year.

Civil Relief includes policies for the rehabilitation of children with disabilities and public welfare funds such as the JD.com Rare Disease Challenge Foundation. Among these, the rehabilitation policy for children with disabilities in Xi’an has a low entry threshold, targeting children aged 0–16 with conditions such as cerebral palsy, intellectual disabilities, autism spectrum disorder, and SMA, providing CNY 28,000 (USD 3920) per child per year. The JD.com Rare Disease Challenge Foundation, which serves patients nationwide, offers an average assistance amount of approximately CNY 20,000 (USD 2800) per case, and applications can be submitted annually. The funds are primarily used to cover the costs of specialized medications for rare diseases like SMA. This fund also has a low entry threshold, with a focus on assisting urban and rural recipients of the minimum living guarantee. The civil relief was also included in the economic burden estimation. In the questionnaires, social relief was included.

Table 2 presents the utilization of healthcare and non-healthcare resources in 2021 for patients with all three types of SMA, detailing specific resources, the number of patients involved, and mean and median values for resource utilization. It is important to note that the hospital visit data only includes complications directly caused by SMA, excluding visits for nusinersen injections, as their costs are incorporated in the medication expenses. For outpatient visits, the mean number per patient in 2021 was 1 (2) for SMA Type 1, 3.05 (3.88) for SMA Type 2, and 2.09 (2.59) for SMA Type 3. Emergency department (ED) visits were relatively low, with an average of 0.25 (0.5), 0.14 (0.48), and 0 (0) for the three SMA types, respectively. Drug usage was limited in 2021 due to availability constraints, with the mean number of drug uses per patient being 0.10 (0.44) for SMA Type 2 and 0.27 (0.90) for SMA Type 3. Each patient had at least one primary caregiver, with an average of 1.33 (0.66) caregivers for SMA Type 2 patients and 1.64 (0.67) caregivers for SMA Type 3 patients. The average daily productivity loss for caregivers was 24 h (0), 19.67 h (7.48), and 14.95 h (9.32) for SMA Types 1, 2, and 3 patients, respectively, reflecting the significant time and resource demands of caregiving. Rehabilitation participation varied, with fewer patients engaging in hospital-based rehabilitation in 2021: one (0.25) for SMA Type 1, ten (47.62%) for SMA Type 2, and five (45.45%) for SMA Type 3. More patients participated in home-based rehabilitation: three (0.75) for SMA Type 1, twenty (95.24%) for SMA Type 2, and six (54.55%) for SMA Type 3. The average number of days spent in hospital-based rehabilitation was highest for SMA Type 2 patients, at 69.10 (97.24) days.

Transportation and accommodation costs were incurred by some patients, particularly among SMA Type 2 patients. For transportation, 57.14% of SMA Type 2 patients experienced costs. Accommodation costs were borne by 33.33% of SMA Type 2 patients. In 2021, 47.62% of SMA Type 2 patients used assistive devices, the highest rate among the three types.

Table 3 shows the use of healthcare and non-healthcare resources in 2022 for all three SMA types, with detailed numbers and mean/median values for resource utilization. There was a marked increase in hospital visits in 2022, with average hospital visits per patient of 1.75 (1.71), 1.62 (2.42), and 1.18 (0.98) for SMA Types 1, 2, and 3, respectively. Outpatient visits remained relatively stable, with mean values of 1 (2), 2.95 (4.02), and 2.82 (3.60) for SMA Types 1, 2, and 3. ED visits remained low, averaging 0 (0), 0.05 (0.22), and 0.09 (0.30) for the three types. Drug usage increased notably due to the availability of nusinersen, though risdiplam remained unavailable. The mean number of nusinersen doses per patient was 2.75 (3.20) for SMA Type 1, 3.57 (2.42) for SMA Type 2, and 3.91 (2.21) for SMA Type 3. The mean number of primary caregivers per patient remained consistent with the previous year: 1 (0) for SMA Type 1, 1.33 (0.66) for SMA Type 2, and 1.64 (0.67) for SMA Type 3. The average daily productivity loss for caregivers was similar to 2021: 24 h (0), 20.8 h (6.0), and 14.95 h (9.3) for SMA Types 1, 2, and 3. Hospital-based rehabilitation participation remained similar to 2021, with the highest average days of rehabilitation seen in SMA Type 2 patients, at 99.6 (102.2) days.

Transportation and accommodation costs were incurred by some patients, with 76.19% of SMA Type 2 and 81.82% of SMA Type 3 patients incurring transportation costs. For accommodation, 47.62% of SMA Type 2 patients and 63.64% of SMA Type 3 patients experienced costs. Assistive devices were used by 57.14% of SMA Type 2 patients, the highest rate among the three types.

Table 4 shows the use of healthcare and non-healthcare resources for 2023. There was a noticeable decrease in hospital visits for SMA Type 3 patients, likely due to the impact of medication. The average hospital visits for SMA Types 1, 2, and 3 were 1 (1.15), 1.14 (2.97), and 0.27 (0.65), respectively. Outpatient visits increased for SMA Type 1 patients, with an average of 4.75 (9.5) visits per patient, compared to 1.05 (1.96) for SMA Type 2 and 1.91 (2.66) for SMA Type 3. ED visits remained low, averaging 0.25 (0.5), 0 (0), and 0.09 (0.30) for the three types. Drug usage of nusinersen decreased, while the use of risdiplam increased, reflecting its convenience. The mean number of nusinersen doses per patient was 2.50 (2.08) for SMA Type 1, 2.14 (1.62) for SMA Type 2, and 2.45 (1.21) for SMA Type 3. The mean number of risdiplam doses per patient was 16.5 (8.58), 13.48 (10.55), and 5.27 (9.06) for SMA Types 1, 2, and 3, respectively. The average daily productivity loss for caregivers was 24 h (0), 20.81 h (6.01), and 15 h (9.69) for SMA Types 1, 2, and 3. Rehabilitation participation remained high for home-based rehabilitation, with SMA Type 1 patients averaging 113.75 (136.5) days of hospital-based rehabilitation, the highest among all types.

For transportation, the number and percentage of patients were 3 (75%) for SMA Type 1, 13 (61.90%) for SMA Type 2, and 10 (90.91%) for SMA Type 3. For accommodation, the number and percentage of patients were one (25%) for SMA Type 1, nine (42.86%) for SMA Type 2, and seven (63.64%) for SMA Type 3.

Figure 1 illustrates the average cost for SMA in 2023. The reason why we chose 2023 for the calculation is because of the accessibility of the medicines. The largest cost is from the productivity loss caused by primary caregivers, accounting for 32.94% of the total cost. Drug usage for nusinersen accounts for 29.29%, followed by drug usage for risdiplam, which accounts for 17.33%. Hospital-based rehabilitation costs account for 8.85%, while hospital visit costs make up 6.99% of the total cost. The direct non-medical costs, including transportation, accommodation, and assistive devices, account for 3.99%. Outpatient costs are relatively low, at only 0.59%, and ED visit costs are so minimal that they are not visible in the figure. Note that the reason we only calculate costs for 2023 is the availability of both nusinersen and risdiplam during that year.

Figure 2 shows the costs for SMA patients in Chinese Yuan (CNY) across 2021, 2022, and 2023. The horizontal axis represents the four SMA types. Bar charts are used to clearly illustrate the costs. In our study, for patients with SMA, excluding hospital rehabilitation costs, expenses across various categories decline as disease severity decreases. The average annual total cost for SMA Type 1 patients was approximately CNY 300,000 (or USD 41,000), for Type 2 patients around CNY 270,000 (or USD 37,000), for Type 3 patients close to CNY 200,000 (or USD 27,000), and for Type 4 patients approximately CNY 80,000 (or USD 11,000) in 2023—significantly lower than the costs for other types. The costs of Disease Modifying Therapies (DMTs) were CNY 145,452 (or USD 19,924.9) for SMA Type 1, CNY 122,147.80 (or USD 16,732.6) for SMA Type 2, CNY 101,414.90 (or USD 13,892.5) for SMA Type 3, and CNY 68,184 (or USD 9340.3) for SMA Type 4. Inpatient costs were CNY 51,970.30 (or USD 7119.2) for SMA Type 1, CNY 18,717.10 (or USD 2564) for Type 2, CNY 4360.67 (or USD 597.4) for Type 3, and CNY 1800 (or USD 246.6) for Type 4. For SMA Type 2 patients, hospital rehabilitation costs were the highest, at CNY 29,100 (or USD 3986.3), followed by CNY 17,700 (or USD 2424.7) for Type 1, CNY 11,850.90 (or USD 1623.4) for Type 3, and CNY 10,000 (or USD 1369.9) for Type 4 patients. Direct non-medical costs were CNY 16,747.50 (or USD 2294.2) for SMA Type 1, CNY 10,253.80 (or USD 1404.6) for Type 2, CNY 7912.73 (or USD 1083.9) for Type 3, and CNY 1950 (or USD 267.1) for Type 4. Productivity loss costs were CNY 91,180 (or USD 12,490.4) for SMA Type 1, CNY 89,823.20 (or USD 12,304.5) for Type 2, CNY 73,565.70 (or USD 10,077.5) for Type 3, and CNY 0 for Type 4.

In 2021, DMT drugs were not included in China’s medical reimbursement list, and a multi-level security system was absent. In 2022, with the inclusion of DMTs in basic medical insurance, a multi-level security framework began to emerge. As shown in Table 5, in 2022, basic medical insurance accounted for 47% of direct costs and 31% of total costs. Commercial insurance reimbursement covered 4% of direct costs and 2% of total costs, while charitable assistance contributed 2% of direct costs and 1% of total costs. Patients’ out-of-pocket expenses accounted for 47% of direct costs and 26% of total costs. In 2023, basic medical insurance reimbursement increased to 49% of direct costs and 32% of total costs. Commercial insurance reimbursement rose to 5% of direct costs and 2% of total costs, while charitable assistance remained at 1% for both direct and total costs. Patient out-of-pocket expenses dropped to 45% of direct costs and 25% of total costs.

## 4. Discussion

Nusinersen was included in the reimbursement list in 2022, and risdiplam was added in 2023 in China. Due to the extremely high costs, patients did not receive adequate treatment in 2021. Additionally, in 2021, China was grappling with COVID-19, and mandatory control measures were in place, making it difficult to visit hospitals. As a result, hospital visits were virtually nonexistent in 2021. Since people were restricted to their homes, the likelihood of acquiring infections leading to SMA-related complications was rare. Furthermore, government control over hospital visits led many patients to avoid hospitals during that period.

In 2022, with the easing of COVID-19 restrictions and the inclusion of nusinersen in the reimbursement list, there was a noticeable increase in hospital visits. The largest use of nusinersen in 2022 was seen among SMA Type 3 patients, followed by SMA Type 2 and SMA Type 1 patients. The average number of hospital visits was highest for SMA Type 1 patients, followed by SMA Type 2 and SMA Type 3 patients.

After risdiplam was included in the reimbursement list in China, it seems that patients began to favor its use over nusinersen, primarily due to its ease of administration. Our findings indicate that, compared to 2022, the overall utilization of nusinersen in patients with various types of SMA has decreased. In 2023, the use of SMA medications appears to have led to a significant reduction in hospital visits. Among the 37 patients in our study, 21 received both nusinersen and risdiplam, as China’s medical insurance does not impose reimbursement restrictions on their combined use. In contrast, commercial insurance policies in some developed countries often limit reimbursement for the combination of drugs for rare diseases [26].

The potentially devastating effects of SMA result in substantial clinical and economic burdens on patients, families, caregivers, and healthcare systems [27]. In 2023, despite the inclusion of both disease-modifying therapies (DMTs) on the reimbursement list in China, six patients (16.21%) still did not receive treatment. This contrasts with the nearly 100% treatment rate in the United States [28]. A study by Droege et al. in the U.S. found that the medical costs for Type 1 SMA patients treated with nusinersen were lower than for those not treated with nusinersen, with annual costs of USD 92,618 versus USD 137,627 per patient, respectively (excluding nusinersen-related costs) [28]. This suggests that treatment with these medicines could reduce overall costs for SMA Type 1 patients. In Canada, a significant proportion of patients were prescribed various SMA treatments, including nusinersen (40.8%, 379/930), risdiplam (33.5%, 312/930), onasemnogene abeparvovec (32.2%, 299/930), and/or olesoxime (18.3%, 170/930) [29].

Except for SMA Type 4 patients, individuals with SMA Types 1, 2, and 3 all require at least one primary caregiver. In China, the parents of children with SMA typically forgo their jobs to assume the caregiving role, although they often lack professional training in this regard. The average caregiving time exceeds 8 h per day, resulting in a significant loss of labor force among the parents of affected children. In contrast, caregivers in Canada reported spending a median of 35 h per week (IQR: 27–55) caring for individuals with SMA [29].

More than half of the SMA patients require hospital-based rehabilitation, with Type 1 patients needing the longest duration, averaging 113.75 (136.5) days. Patients with SMA Types 2 and 3 require an average of 84.76 (96.84) and 80.09 (101.28) days, respectively, for hospital rehabilitation. Additionally, most patients require home-based rehabilitation, which is often carried out by family members without professional training, with rehabilitation durations exceeding 8 h per day. In contrast, in developed countries, professional caregivers are typically employed to assist children with SMA. In Canada, 46.5% of caregivers (442/951) reported receiving paid support from professional caregivers [29].

In this study, we primarily focus on the cost distribution in 2023. Regarding direct medical costs, we categorized SMA-related costs into hospital visits, outpatient visits, medication, and hospital-based rehabilitation. Similarly to other studies, direct medical costs in our research also include hospital visits, outpatient visits, medications, emergency services, and in-hospital rehabilitation. The primary economic burden of direct medical costs for SMA patients in these countries arises from hospitalization expenses, while data on DMT (disease-modifying treatment) costs remain limited.

For patients receiving nusinersen, medication costs are the highest, surpassing those of hospitalization and rehabilitation. A study based on claims data indicates that the total annualized cost for the nusinersen treatment subgroup averages USD 792,263, with the cost of nusinersen treatment alone accounting for USD 717,943, or nearly 90% of direct medical costs in the U.S. [30]. Our study reveals that DMT costs account for 75.38% of direct medical expenses in 2023. Consistent with international studies, our findings show that the economic burden of SMA is closely related to the type of SMA, with SMA Type 1 incurring the highest costs, followed by Types 2, 3, and 4 [19]. Regarding direct non-medical costs, SMA patients in our study incurred the necessary expenses for transportation, accommodation, and medical devices, consistent with international findings [21].

The productivity losses resulting from caregiving are also significant. As shown in Figure 1, productivity loss accounted for approximately 33.35% of the total cost in our study. In China, caregivers are more likely to become unemployed due to the demands of caregiving. In contrast, caregivers in some countries typically remain employed, though they may experience absenteeism or reduced work hours [31].

Although medical insurance reimbursed CNY 95,353 (or USD 13,062.1), patients still had to pay CNY 74,152 (or USD 10,157.8) out of pocket, which accounted for 25.73% of the total cost. Notably, patients’ indirect costs, such as productivity loss, amounted to CNY 82,709 (or USD 11,330). Given China’s current development level, with a per capita GDP of CNY 89,360 (or USD 12,241.1) in 2023, this was a substantial financial burden for patients’ families.

Basic medical insurance plays a vital role as the cornerstone of patient security, especially in terms of reimbursement. However, commercial insurance has a relatively limited role in covering the costs for SMA patients. While some commercial insurance products may cover treatment for certain rare diseases, their coverage is narrow, and reimbursement rates are generally low. Charitable assistance also provides support to Chinese SMA patients, with some organizations and foundations offering financial aid and drug donations. However, the funding sources for these charitable efforts are unstable, and their coverage is relatively limited, making it difficult to meet the needs of all patients.

Indirect medical costs continue to account for a significant portion of the total expenses. We believe that Long-Term Care Insurance (LTCI) could play a critical role in addressing the financial and caregiving burdens associated with SMA. By providing financial security, reducing caregiving demands, and improving access to quality care, LTCI would not only benefit patients and families but also contribute to a more sustainable and efficient healthcare system. Given China’s rapidly aging population and the rising prevalence of chronic diseases, the establishment of an LTCI system has become a crucial policy priority. The country’s efforts to develop LTCI reflect its commitment to addressing the growing demand for long-term care services and alleviating the associated financial and caregiving pressures on individuals and families.

Conducting a cost-effectiveness analysis (CEA) for Spinal Muscular Atrophy (SMA) can provide numerous benefits, both for healthcare systems and for patients, as well as for policymakers and pharmaceutical companies. CEA offers a range of benefits for the treatment of SMA, including improved resource allocation, informed decision-making, sustainability of healthcare systems, and promoting equity in access to care. By systematically evaluating the costs and benefits of SMA therapies, CEA helps ensure that resources are used efficiently and that patients receive the most effective treatments at a sustainable cost. As the availability of SMA therapies continues to expand, conducting CEA will be crucial in shaping policies that balance innovation with the practical realities of healthcare funding. In order to measure health-related quality of life to support economic evaluations in SMA population, Richard. H. Xu et al. conducted measurement properties of the EQ-5D-5L and Patient-Reported Outcomes Measure Information System Preference measure (PROPr) in patients with SMA [32]. Strong ceiling and floor effects were observed for four dimensions of the EQ-5D-5L and three subscales, including pain intensity, pain interference, and physical function, of the PROMIS-29. The same group also conducted the measurement properties of the EQ-5D-Y-3L, PedsQL 4.0, and PROMIS-25 Profile v2.0 in pediatric patients with SMA [33].

A key limitation of this study is the retrospective nature of the data collection, which introduces the potential for recall bias. Nevertheless, we examined the utilization of both nusinersen and risdiplam. Since nusinersen is administered via intrathecal injection and has been in use for only two years (with a standard dosage of six injections in the first year, followed by a maintenance dose of four times per year), the recollections of patients’ parents are likely to be relatively accurate. Additionally, we incorporated data from the Shaanxi Provincial Healthcare Security Bureau to conduct relevant analyses on nusinersen use, which enhances the robustness of our survey results (see Appendix A).

## 5. Conclusions

This study highlights that the integration of DMT into basic medical insurance is crucial for meeting the treatment needs of Chinese patients with SMA. The high cost of DMTs, combined with the loss of labor from primary caregivers, creates a significant financial burden on both SMA patients and the healthcare system in China. While basic medical insurance serves as a foundational pillar for patient support, the multi-level security framework for healthcare remains incomplete. Developing a comprehensive multi-level security system is essential for addressing the unique challenges posed by rare diseases and warrants further research and policy development.

## Figures and Tables

**Figure 1 healthcare-13-00428-f001:**
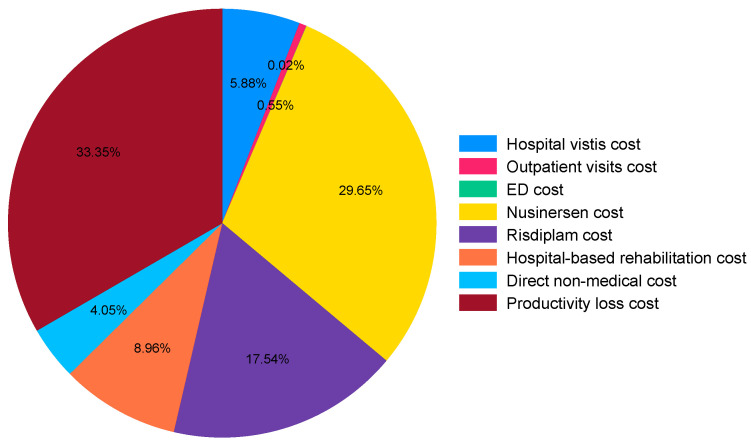
The percentage of cost in 2023.

**Figure 2 healthcare-13-00428-f002:**
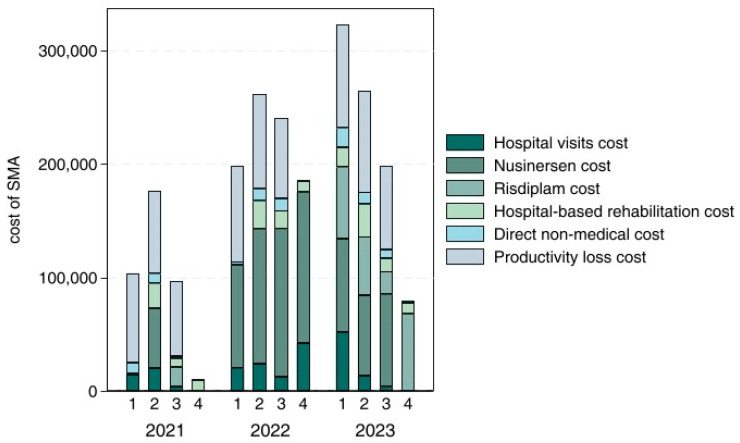
The cost for the SMA patients in the unit of Chinese Yuan (CNY 1 = USD 0.14).

**Table 1 healthcare-13-00428-t001:** Demographic and clinical characteristics of SMA study participants (*n* = 37 patient-caregiver pairs).

	SMA 1	SMA 2	SMA 3	SMA 4	Total
**Patient details**					
Current age, mean (SD), y	5.1 (4.8)	9.4 (3.9)	11.5 (10.1)	25.0 (0.0)	10.0 (7.0)
Sex, *n* (%)					
Male	2 (50.0%)	12 (57.1%)	5 (45.5%)	1 (100.0%)	20 (54.1%)
Female	2 (50.0%)	9 (42.9%)	6 (54.5%)	0 (0.0%)	17 (45.9%)
Registered permanent residence #, *n* (%)					
Urban	1 (25.0%)	14 (66.7%)	3 (27.3%)	0 (0.0%)	18 (48.6%)
Rural	3 (75.0%)	7 (33.3%)	8 (72.7%)	1 (100.0%)	19 (51.4%)
Disability, *n* (%)					
Yes	3 (75.0%)	20 (95.2%)	7 (63.6%)	0 (0.0%)	30 (81.1%)
No	1 (25.0%)	1 (4.8%)	4 (36.4%)	1 (100.0%)	7 (18.9%)
Disability level, *n* (%)					
Grade 1 Disability	3 (100.0%)	12 (60.0%)	2 (28.6%)	0 (0.0%)	17 (56.7%)
Grade 2 Disability	0 (0.0%)	7 (35.0%)	2 (28.6%)	0 (0.0%)	9 (30.0%)
Grade 3 Disability	0 (0.0%)	0 (0.0%)	3 (42.8%)	0 (0.0%)	3 (10.0%)
Grade 4 Disability	0 (0.0%)	0 (0.0%)	0 (0.0%)	0 (0.0%)	0 (0.0%)
Grade 5 Disability	0 (0.0%)	1 (5.0%)	0 (0.0%)	0 (0.0%)	1 (3.2%)
Commercial insurance, *n* (%)					
Yes	4 (100.0%)	16 (76.2%)	10 (90.9%)	0 (0.0%)	30 (81.1%)
No	0 (0.0%)	5 (23.8%)	1 (9.1%)	1 (100.0%)	7 (18.9%)
Inclusive commercial insurance, *n* (%)					
Yes	4 (100.0%)	14 (66.7%)	10 (90.9%)	0 (0.0%)	28 (75.7%)
No	0 (0.0%)	7 (33.3%)	1 (9.1%)	1 (100.00%)	9 (24.3%)
Civil relief, *n* (%)					
Yes	1 (25.0%)	8 (38.1%)	4 (36.4%)	1 (100.0%)	14 (37.8%)
No	3 (75.0%)	13 (61.9%)	7 (63.6%)	0 (0.0%)	23 (62.1%)
Social assistance, *n* (%)					
Yes	2 (50.0%)	9 (42.9%)	2 (18.1%)	0 (0.0%)	13 (35.1%)
No	2 (50.0%)	12 (57.1%)	9 (81.9%)	1 (100.0%)	24 (64.9%)
**Current disease state**					
Age of symptom onset, mean (SD), y	0.3 (0.2)	0.9 (0.4)	3.2 (3.7)	19.0 (0.0)	2.0 (3.7)
Age at diagnosis, mean (SD), y	0.6 (0.2)	1.8(1.3)	6.7 (8.5)	20.0 (0.0)	3.6 (5.9)
Scoliosis, *n* (%)					
Yes	1 (25.0%)	19 (90.5%)	6 (54.5%)	0 (0.0%)	26 (70.3%)
No	3 (75.0%)	2 (9.5%)	5 (45.5%)	1 (100.0%)	11 (29.7%)
Number of hospitals visited before diagnosis, median (range)	1.8 (1–2)	3.9 (1–13)	2.1 (0–10)	1.0 (1–1)	4.0 (1–13)
**Caregiver and family**					
Household size, median (range)	3.3 (2–4)	4.8 (2–7)	4.1 (2–5)	3.0 (3–3)	4.4 (2–7)
Marital status, *n* (%)					
Married	2 (50.0%)	21 (100.0%)	9 (81.8%)	1 (100.0%)	33 (89.2%)
Divorced	2 (50.0%)	0 (0.0%)	2 (18.1%)	0 (0.0%)	4 (10.8%)
Number of caregivers, *n* (%)					
0	0 (0.0%)	0 (0.0%)	0 (0.0%)	1 (100.0%)	1 (2.7%)
1	4 (100.0%)	8 (38.1%)	4 (36.4%)	0 (0.0%)	16 (43.2%)
2	0 (0.0%)	6 (28.6%)	6 (54.5%)	0 (0.0%)	12 (32.4%)
3	0 (0.0%)	3 (14.3%)	0 (0.0%)	0 (0.0%)	3 (8.1%)
4	0 (0.0%)	4 (19.0%)	1 (9.1%)	0 (0.0%)	5 (13.5%)
Primary caregiver’s age, mean (SD), y	36.3 (5.0)	40.1 (5.8)	36.5 (5.1)	25.0 (0.0)	38.1 (6.0)
Primary caregiver’s sex, *n* (%)					
Male	0 (0.0%)	1 (4.8%)	2 (18.2%)	1 (100.0%)	4 (10.8%)
Female	4 (100.0%)	20 (95.2%)	9 (81.8%)	0 (0.0%)	33 (89.2%)
Primary caregiver’s employment status, *n* (%)					
Employed full-time	0 (0.0%)	4 (19.0%)	1 (9.1%)	0 (0.0%)	5 (13.5%)
Employed part-time	0 (0.0%)	1 (4.8%)	3 (27.3%)	1 (100.0%)	5 (13.5%)
Unemployed	4 (100.0%)	16 (76.2%)	7 (63.6%)	0 (0.0%)	27 (73.0%)
Duration of caregiving for the SMA child per day, median (range), h	24.0 (24.0–24.0)	20.8 (5.5–24.0)	15.0 (3.0–24.0)	0.0 (0.0–0.0)	19.0 (3.0–24.0)
Family income, *n* (%)					
RMB 10,000 to 60,000	3 (75.0%)	13 (61.9%)	6 (54.5%)	1 (100.0%)	23 (62.2%)
RMB 70,000 to 90,000	1 (25.0%)	4 (19.0%)	4 (36.4%)	0 (0.0%)	9 (24.3%)
RMB 100,000 to 130,000	0 (0.0%)	3 (14.3%)	1 (9.1%)	0 (0.0%)	4 (10.8%)
RMB 140,000 to 170,000	0 (0.0%)	0 (0.0%)	0 (0.0%)	0 (0.0%)	0 (0.0%)
RMB 180,000 to 210,000	0 (0.0%)	0 (0.0%)	0 (0.0%)	0 (0.0%)	0 (0.0%)
RMB More than 220,000	0 (0.0%)	1 (0.0%)	0 (0.0%)	0 (0.0%)	1 (2.7%)
Father education, *n* (%)					
Junior high school or below	1 (25.0%)	5 (23.8%)	5 (45.5%)	0 (0.0%)	11 (29.7%)
Senior high school	2 (50.0%)	4 (19.0%)	2 (18.2%)	1 (100.0%)	9 (24.3%)
College/university or above	1 (25.0%)	12 (57.1%)	4 (36.4%)	0 (0.0%)	17 (45.9%)
Mother education, *n* (%)					
Junior high school or below	1 (25.0%)	5 (23.8%)	6 (54.5%)	0 (0.0%)	12 (32.4%)
Senior high school	0 (0.0%)	5 (23.8%)	1 (9.1%)	1 (100.0%)	7 (18.9%)
College/university or above	3 (75.0%)	11 (52.4%)	4 (36.4%)	0 (0.0%)	18 (48.6%)

Note: # this is a household registration system that categorizes individuals based on their place of registration (residence) and their classification as urban (non-agricultural) or rural (agricultural) residents. The urban residents are typically registered in cities or towns. They have access to social benefits available in cities, such as urban healthcare and urban public education. Rural residents are registered in villages or rural regions traditionally linked to agricultural activities and farming. Rural resident holders often have limited access to urban benefits but may have land-use rights for farming in their registered village.

**Table 2 healthcare-13-00428-t002:** Resource use per patient/year for all three types of patients in 2021.

Variable	SMA 1		SMA 2		SMA 3	
	Mean (SD)	Median (IQR)	Mean (SD)	Median (IQR)	Mean (SD)	Median (IQR)
Hospital visits #	0 (0)	0 (0–0)	0 (0)	0 (0–0)	0 (0)	0 (0–0)
Outpatient visits #	1 (2)	0 (0–2)	3.05 (3.88)	2 (0–5)	2.09 (2.59)	2 (0–3)
ED visits	0.25 (0.5)	0 (0–0.5)	0.14 (0.48)	0 (0–0)	0 (0)	0 (0–0)
Drugs (number of ampoule/bottle)						
Nusinersen	0 (0)	0 (0–0)	0.10 (0.44)	0 (0–0)	0 (0)	0 (0–0)
Risdiplam	0 (0)	0 (0–0)	0 (0)	0 (0–0)	0.27 (0.90)	0 (0–0)
Number of primary caregivers	1 (0)	1 (1–1)	1.33 (0.66)	1 (1–1)	1.64 (0.67)	2 (1–2)
	Utilization, *n* (%)		Utilization, *n* (%)		Utilization, *n* (%)	
Hospital-based rehabilitation	1 (25%)		10 (47.62%)		5 (45.45%)	
Home-based rehabilitation	3 (75%)		20 (95.24%)		6 (54.55%)	
Assistive devices	2 (50%)		10 (47.62%)		5 (45.45%)	
Transport	1 (25%)		12 (57.14%)		4 (36.36%)	
Accommodation	2 (50%)		7 (33.33%)		3 (27.27%)	
	Utilization, days per year		Utilization, days per year		Utilization, days per year	
Hospital-based rehabilitation	7.5 (15)	0 (0–15)	69.10 (97.24)	0 (0–108)	46.09 (78.07)	6 (0–60)
	Utilization, hours per day		Utilization, hours per day		Utilization, hours per day	
Productivity loss	24 (0)	24 (24–24)	19.67 (7.48)	24 (13.5–24)	14.95 (9.32)	15 (5–24)

Note: # represents that the resource utilization of hospital visits as well as outpatient visits is related to the resource utilization of SMA-related complications, and has nothing to do with DMT.

**Table 3 healthcare-13-00428-t003:** Resource use per patient/year for all three types of patients in 2022.

Variable	SMA 1		SMA 2		SMA 3	
	Mean (SD)	Median (IQR)	Mean (SD)	Median (IQR)	Mean (SD)	Median (IQR)
Hospital visits #	1.75 (1.71)	1.5 (0.5–3)	1.62 (2.42)	0 (0–2)	1.18 (0.98)	1 (1–1)
Outpatient visits #	1 (2)	0 (0–2)	2.95 (4.02)	0 (0–5)	2.82 (3.60)	2 (0–6)
ED visits	0 (0)	0 (0–0)	0.05 (0.22)	0 (0–0)	0.09 (0.30)	0 (0–0)
Drugs (number of ampoule/bottle)						
Nusinersen	2.75 (3.20)	2.5 (0–5.5)	3.57 (2.42)	5 (0–5)	3.91 (2.21)	5 (2–5)
Risdiplam	0 (0)	0 (0–0)	0 (0)	0 (0–0)	0 (0)	0 (0–0)
Number of primary caregivers	1 (0)	1 (1–1)	1.33 (0.66)	1 (1–1)	1.64 (0.67)	2 (1–2)
	Utilization, *n* (%)		Utilization, *n* (%)		Utilization, *n* (%)	
Hospital-based rehabilitation	0 (0%)		12 (57.14%)		5 (45.45%)	
Home-based rehabilitation	3 (75%)		20 (95.24%)		6 (54.55%)	
Assistive devices	0 (0%)		12 (57.14%)		4 (36.36%)	
Transport	3 (75%)		16 (76.19%)		9 (81.82%)	
Accommodation	1 (25%)		10 (47.62%)		7 (63.64%)	
	Utilization, days per year		Utilization, days per year		Utilization, days per year	
Hospital-based rehabilitation	0 (0)	0 (0–0)	99.6 (102.2)	108 (0–195)	66.5 (100.2)	28 (0–100)
	Utilization, hours per day		Utilization, hours per day		Utilization, hours per day	
Productivity loss	24 (0)	24 (24–24)	20.8 (6.0)	24 (24–24)	14.9 (9.3)	15 (5–24)

Note: # represents that the resource utilization of hospital visits as well as outpatient visits is related to the resource utilization of SMA-related complications, and has nothing to do with DMT.

**Table 4 healthcare-13-00428-t004:** Resource use per patient/year for all three types of patients in 2023.

Variable	SMA 1		SMA 2		SMA 3	
	Mean (SD)	Median (IQR)	Mean (SD)	Median (IQR)	Mean (SD)	Median (IQR)
Hospital visits #	1 (1.15)	1 (0–2)	1.14 (2.97)	0 (0–0)	0.27 (0.65)	0 (0–0)
Outpatient visits #	4.75 (9.5)	0 (0–9.5)	1.05 (1.96)	0 (0–0)	1.91 (2.66)	0 (0–3)
ED visits	0.25 (0.5)	0 (0–0.5)	0 (0)	0 (0–0)	0.09 (0.30)	0 (0–0)
Drugs (number of ampoule/bottle)						
Nusinersen	2.50 (2.08)	2.5 (1–4)	2.14 (1.62)	3 (0–3)	2.45 (1.21)	3 (3–3)
Risdiplam	16.5 (8.58)	18.5 (10–23)	13.48 (10.55)	17 (0–20)	5.27 (9.06)	0 (0–15)
Number of primary caregivers	1 (0)	1 (1–1)	1.33 (0.66)	1 (1–1)	1.64 (0.67)	2 (1–2)
	Utilization, *n* (%)		Utilization, *n* (%)		Utilization, *n* (%)	
Hospital-based rehabilitation	2 (50%)		11 (52.38%)		6 (54.55%)	
Home-based rehabilitation	3 (75%)		20 (95.24%)		6 (54.55%)	
Assistive devices	2 (50%)		11 (52.38%)		3 (27.27%)	
Transport	3 (75%)		13 (61.90%)		10 (90.91%)	
Accommodation	1 (25%)		9 (42.86%)		7 (63.64%)	
	Utilization, days per year		Utilization, days per year		Utilization, days per year	
Hospital-based rehabilitation	113.75 (136.5)	91 (0–227.5)	84.76 (96.84)	48 (0–154)	80.09 (101.28)	56 (0–119)
	Utilization, hours per day		Utilization, hours per day		Utilization, hours per day	
Productivity loss	24 (0)	24 (24–24)	20.81 (6.01)	24 (24–24)	15 (9.69)	18 (5–24)

Note: # represents that the resource utilization of hospital visits as well as outpatient visits is related to the resource utilization of SMA-related complications, and has nothing to do with DMT.

**Table 5 healthcare-13-00428-t005:** Costs of patients with SMA in CNY and multi-level insurance coverage rates.

	2021		2022		2023	
	Mean (SD)	Median (IQR)	Mean (SD)	Median (IQR)	Mean (SD)	Median (IQR)
Total cost #	153,367.00 (190,372.1)	92,320 (80,520–92,320)	252,688.40 (108,736)	272,456 (152,704–333,834)	252,214.30 (115,484.3)	247,281 (177,627.5–298,632)
Direct cost	83,600.92 (187,965.70)	19,842 (5600–80,735)	175,468.10 (106,076)	193,280 (68,360–259,559)	169,505.40 (109,071.5)	166,436 (95,972–221,741)
Product loss cost *	69,766.08 (20,760.75)	78,520 (78,520–78,520)	77,220.35 (18,396.46)	84,344 (84,344–84,344)	82,708.88 (21,850.25)	91,180 (91,180–91,180)
Multi-tiered insurance	7810.81 (20,072.24)	0 (0–3600)	96,800.97 (62,159.78)	113,280 (28,900–140,800)	95,352.99 (59,907.69)	101,843.3 (57,929.6–134,587.2)
Patient out of pocket	75,790.11 (185,062.50)	19,842 (5000–64,000)	78,667.12 (60,363.76)	65,000 (35,801–122,096)	74,152.46 (65,134.07)	58,082.4 (32,498–99,104)
	Direct cost	Total cost	Direct cost	Total cost	Direct cost	Total cost
Proportion Covered by Multi-level Medical Security	0.09 (0.16)	0.04 (0.08)	0.53 (0.25)	0.34 (0.19)	0.55 (0.28)	0.35 (0.18)
Proportion Covered by First-level Medical Security	0.08 (0.16)	0.03 (0.08)	0.47 (0.23)	0.31 (0.19)	0.49 (0.23)	0.32 (0.16)
Proportion Covered by Second-level Medical Security	0.00 (0.02)	0.00 (0.01)	0.04 (0.14)	0.02 (0.05)	0.05 (0.11)	0.02 (0.04)
Proportion Covered by Third-level Medical Security	0.00 (0.01)	0.00 (0.00)	0.02 (0.10)	0.01 (0.04)	0.01 (0.04)	0.01 (0.03)
Proportion Covered by Patient out of pocket	0.91 (0.16)	0.30 (0.28)	0.47 (0.25)	0.26 (0.17)	0.45 (0.28)	0.26 (0.16)

Note: # the cost is shown in units of CNY. The exchange rate between CNY and USD in 2025 is CNY 1 = USD 0.14. * The cost of individual productivity loss is calculated based on the per capita annual income in Shaanxi Province.

## Data Availability

The data presented in this study are not available due to privacy concerns.

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
