# Peer review of "Healthcare Resource Utilization, Economic Burden, and Multi-Level Medical Security System for Individuals with Spinal Muscular Atrophy in Shaanxi Province, China"

_healthcare, 2025, doi:10.3390/healthcare13040428_

Round 1
Reviewer 1 Report
Comments and Suggestions for Authors
Abstract: Body is over 330 words. Recommend closer to 250 as recommended in instruction for authors.
Introduction: Would recommend selling out all abbreviations in the introduction as if they were not presented in the abstract.
Results: The tables are nicely put together, but the text of the results is reporting too much of what is already in the tables. Consider being more concise and emphasize most important aspects of the data only.
Would recommend commenting on the meaning of the results in the discussion section and just reporting the results within this section. (Line 210)
Author Response
REVIEWER 1
1.Abstract: Body is over 330 words. Recommend closer to 250 as recommended in instruction for authors.
Responses: Thanks for your comments. We agree the reviewer’s comments that there are too many words in the abstract. We have reduced the comments’ words. Here are some sentences which are not so important in the abstract section. We have removed them:
Data were collected from SMA patients recruited through the Meier Advocacy and Support Center (MASC) for SMA.
A total of 37 patient-caregiver pairs, with an average patient age of 10 years, participated in the study.
Healthcare costs were calculated based on actual healthcare service utilization by each patient from 2021 to 2023.
Prior to the inclusion of nusinersen and risdiplam on China's National Reimbursement Drug List (NRDL), SMA patients in China lacked access to adequate treatment.
2.Introduction: Would recommend selling out all abbreviations in the introduction as if they were not presented in the abstract.
Responses: Thanks for your comments and thanks for your careful check of our manuscript. We have checked again all the abbreviations in our manuscript and we have marked them into red in the manuscript when they were changed. For example, we have added National Reimbursement Drug List(NRDL) in the abstract section.
3.Results: The tables are nicely put together, but the text of the results is reporting too much of what is already in the tables. Consider being more concise and emphasize most important aspects of the data only.
Would recommend commenting on the meaning of the results in the discussion section and just reporting the results within this section. (Line 210)
Responses: Thanks for your comments. We have read the manuscript again and removed some sentences such as:
“, and the distribution between rural (51.4%) and urban (48.6%) residents was nearly equal”
“Civil relief was provided to 37.8% of the patients, while 35.1% received social assistance.”
“Over 62% of families had an annual income below ¥60,000, and among the parents, 29.7% of fathers and 32.4% of mothers had a junior high school education level.”
For the description of table 2, we have removed some sentences:
“There were no hospital visits in 2021 for any patient type.”
For description of table 4, we have removed some sentences:
“Transportation and accommodation costs were not incurred by all patients.”
“In 2023, not all patients required assistive devices. The number of patients who needed assistive devices and the corresponding percentages were 2 (50%) for SMA Type 1, 11 (52.38%) for SMA Type 2, and 3 (27.27%) for SMA Type 3.”

Reviewer 2 Report
Comments and Suggestions for Authors
The authors conducted an observational study to quantify healthcare resource utilization, economic burden, and the multi-level medical security system for Spinal Muscular Atrophy (SMA). Participants were enrolled via an online survey recruited through the Meier Advocacy Support Center (MASC) for SMA. The retrospective data analysis was employed to calculate the healthcare cost and resource utilization by each patient from 2021 to 2023.
Overall, this is a well-written article. I have a couple of comments:
- In the abstract, the author concluded that commercieal insurance has relatively limited impact on covering the costs for SMA patients. However, it is not clear which results in the abstract connect to this statement. Although the commercial insurance coverage was discussed in the main text, the author should rephrase this statement based on the information presented.
- Different currency was used to present the healthcare cost. I would suggest authors add the information of USD so that international readers can easily follow.
- 37 patient-caregiver pairs were recruited via the online survey by MASC, it is not clear whether this approach could ensure the representativeness of the participants compared to general SMA population. The authors should discuss the advantage of using this approach and interpretation whether the findings can be generalizable.
- In table 1, how the registered permanent regidence (Urban, Rural) was defined?
Thank you for having me review this article.
Author Response
Responses to the reviewers:
REVIEWER 2
The authors conducted an observational study to quantify healthcare resource utilization, economic burden, and the multi-level medical security system for Spinal Muscular Atrophy (SMA). Participants were enrolled via an online survey recruited through the Meier Advocacy Support Center (MASC) for SMA. The retrospective data analysis was employed to calculate the healthcare cost and resource utilization by each patient from 2021 to 2023.
Overall, this is a well-written article. I have a couple of comments:
- In the abstract, the author concluded that commercial insurance has relatively limited impact on covering the costs for SMA patients. However, it is not clear which results in the abstract connect to this statement. Although the commercial insurance coverage was discussed in the main text, the author should rephrase this statement based on the information presented.
Responses: Thanks for your comments. This is really a good question. We have added the following sentence in the abstract:
“Patients still had to pay 25.73% of the total cost for the direct costs.”
Due to the limited words in the abstract, we could not add a lot of sentences in the abstract. The sentence we added imply that patients need to pay 25.73% of the total cost which is not covered by the basic medical insurance. This is still a lot for the patients. So, the commercial insurance could play a key role for the residual payment of the medical cost. This is a good solution to this problem and thus we conclude that the commercial insurance has a relatively limited impact on covering the costs.
2.Different currency was used to present the healthcare cost. I would suggest authors add the information of USD so that international readers can easily follow.
Responses: Thanks for your comments. This is a good question. We have added the equivalence USD data in the manuscript. Here is an example of the correction:
“Although medical insurance reimbursed 95,353 yuan(or 13,062.1 USD), patients still had to pay 74,152 yuan(or 10,157.8 USD) out-of-pocket, which accounted for 25.73% of the total cost. Notably, patients' indirect costs, such as productivity loss, amounted to 82,709 yuan(or 1,1330 USD). Given China's current development level, with a per capita GDP of 89,360yuan(or 12,241.1 USD) in 2023, this remains a substantial financial burden for patients' families.”
- 37 patient-caregiver pairs were recruited via the online survey by MASC, it is not clear whether this approach could ensure the representativeness of the participants compared to general SMA population. The authors should discuss the advantage of using this approach and interpretation whether the findings can be generalizable.
Responses: Thanks for your comments. This is a good question. We believe that our study represents the general SMA population in China. First, the cost of nusinersen and risdiplam, as well as the percentage covered by the reimbursement policy in China, is consistent nationwide. Second, China exhibits significant variation in GDP development across provinces. We selected Shaanxi Province as a representative region, as it falls within the middle range of GDP development in China. Third, acquiring a sample of SMA patients is challenging. Therefore, we are confident that our study reflects the general SMA population.
4.In table 1, how the registered permanent regidence (Urban, Rural) was defined?
Responsese:It is a household registration system that categorizes individuals based on their place of registration (residence) and their classification as urban (non-agricultural) or rural (agricultural) residents. The urban residents are typically registered in cities or towns.They access to social benefits available in cities, such as: Urban healthcare and Urban public education.
Rural residents are registered in villages or rural regions. Traditionally linked to agricultural activities and farming. Rural residents holders often have limited access to urban benefits but may have land-use rights for farming in their registered village.
We have added this at the bottom of table.1 as a note.

Reviewer 3 Report
Comments and Suggestions for Authors
The paper is well-written. Although the sample size is only 37, the dyadic design could provide useful evidence for understanding the impact of SMA on patients' health.
- Authors should provide more information about SMA in China, including treatment approaches, relevant policies, and their impacts.
- More details about patient recruitment should be provided, as most readers may not be able to read Chinese sources.
- There is a discrepancy in the data collection timeline. Authors state that "Data were primarily collected through resource use questionnaires completed by parents (typically the mother) of 36 children diagnosed with SMA Types I, II, and III, as well as one adult patient in China, between 2021 and 2023." However, they later indicate that "The survey was available online for one month, from July 2024 to October 2024." Which description is correct?
- The use of descriptive analysis alone is insufficient. We need to understand whether the observed changes are statistically meaningful.
- While the study collected data from 37 patient-caregiver dyads, there is no analysis based on this dyadic relationship, which diminishes the value of this study design.
- In the discussion, authors should address the potential benefits of conducting cost-effectiveness analysis for SMA. Given that EQ-5D showed good performance in both adult and pediatric populations, further CEA is warranted. Please consider citing these references to support this suggestion (10.1186/s12955-023-02204-z and 10.1186/s12955-024-02264-9).
- I recommend the authors revise the discussion thoroughly and maintain a more modest tone given the small sample size of 37, which may introduce significant bias. This study appears to be a pilot, and such strong statements should be reserved for more comprehensive formal studies.
Author Response
Responses to the reviewers:
REVIEWER3
The paper is well-written. Although the sample size is only 37, the dyadic design could provide useful evidence for understanding the impact of SMA on patients' health.
- Authors should provide more information about SMA in China, including treatment approaches, relevant policies, and their impacts.
Responses: Thanks for your comments. We have added more talks about the treatment approaches, relevant policies in the second paragraph of the introduction section. We show them again here:
The drugs for treating SMA include Nusinersen and Risdiplam. Before 2022, although both drugs were available on the market, they were not included in the reimbursement program, and the number of patients using them was very low. Through national drug price negotiations (In 2018, China introduced a policy called Volume-based Procurement (VBP), which significantly impacted the price negotiation process. Under VBP, the government conducts bulk procurement of drugs through a competitive bidding process. The goal is to reduce drug prices through competition.), nusinersen and risdiplam were included in China’s NRDL in 2022 and 2023, respectively [14,15]. Furthermore, China’s latest medical security plan, outlined by the General Office of the State Council, introduces a multi-tiered medical insurance system aimed at benefiting all citizens, including those with rare diseases [16]. This system consists of three main levels: basic medical insurance, supplementary medical insurance, and social assistance programs. Basic medical insurance includes both employee and resident medical insurance, critical illness insurance, and medical assistance programs (see Supplementary Table 1) [17]. After being included in the health insurance system, most patients are now able to receive treatment.
- More details about patient recruitment should be provided, as most readers may not be able to read Chinese sources.
Responses: Thanks for your comments. We have added more details in section 2.2. We show it again here:
SMA patients and their caregivers were recruited through the Meier Advocacy and Support Center(MASC) for SMA [25], the only professional nonprofit organization specializing in SMA in China. Participants were eligible for inclusion in this study if they were family members of SMA patients or adult patients who had knowledge of their own treatment and associated costs, demonstrated good compliance, and either resided in or received treatment in Shanxi Province, China. Patients involved in long-term interventional clinical trials (including extension studies) were excluded from the study. If a patient was unable to read or write, their caregiver was allowed to complete the survey on their behalf. However, caregivers who could not read or write were excluded. An invitation letter, a detailed study information sheet, and consent forms were distributed to all eligible participants. We contacted MASC’s principal in Shanxi Province via email and posted a recruitment notice in WeChat groups, explaining the study's purpose and significance. A total of 42 families were invited to participate through voluntary sign-ups and recommendations from the principal. However, 5 families withdrew prior to the interviews due to concerns about the privacy of the recording process. As a result, we received 37 valid responses. To validate the sample size, we consulted the Shanxi Provincial Medical Insurance Database, which reported 34 patients using Nusinersen in 2022, aligning closely with the 27 patients from our study that year. Among the 37 respondents, 3 were adult patients, and the remaining 34 were family members (primary caregivers). All participants provided verbal informed consent and received compensation based on the interview duration.
- There is a discrepancy in the data collection timeline. Authors state that "Data were primarily collected through resource use questionnaires completed by parents (typically the mother) of 36 children diagnosed with SMA Types I, II, and III, as well as one adult patient in China, between 2021 and 2023." However, they later indicate that "The survey was available online for one month, from July 2024 to October 2024." Which description is correct?
Responses: Thanks for your comments. We are sorry for the naïve mistake. Actually, these data are related to resource utilization for SMA patients between 2021 and 2023. The survey was conducted in October 2024. We have corrected it in the data collection section.
- The use of descriptive analysis alone is insufficient. We need to understand whether the observed changes are statistically meaningful.
Responses: Thanks for your comments. We use descriptive statistics (mean and standard deviation [SD] for continuous variables; count and proportion for categorical variables) were used to describe the demographic and clinical characteristics of the SMA population by disease type. Due to the skewed distribution of resource use and costs, both the mean [SD] and median [IQR] values were reported. Data were entered using EpiData with double data entry and cross-checking. All analyses were performed using STATA 16.
- While the study collected data from 37 patient-caregiver dyads, there is no analysis based on this dyadic relationship, which diminishes the value of this study design.
Responses: Thanks for your comments. We have added more in section 2.2. We show them again here:
Participants were eligible for inclusion in this study if they were family members of SMA patients or adult patients who had knowledge of their own treatment and associated costs, demonstrated good compliance, and either resided in or received treatment in Shanxi Province, China. Patients involved in long-term interventional clinical trials (including extension studies) were excluded from the study. If a patient was unable to read or write, their caregiver was allowed to complete the survey on their behalf. However, caregivers who could not read or write were excluded.
We think that we have talked more about the patient-caregiver relationship.
- In the discussion, authors should address the potential benefits of conducting cost-effectiveness analysis for SMA. Given that EQ-5D showed good performance in both adult and pediatric populations, further CEA is warranted. Please consider citing these references to support this suggestion (10.1186/s12955-023-02204-z and 10.1186/s12955-024-02264-9).
Responses: Thanks for your comments. We really appreciate for your comments. Addressing the benefits of conducting cost-effectiveness analysis for SMA is a good suggestion. Conducting cost-effectiveness analysis is really important for SMA. We have discussed this in the discussion section. We show it again here:
Conducting a cost-effectiveness analysis (CEA) for Spinal Muscular Atrophy (SMA) can provide numerous benefits, both for healthcare systems and for patients, as well as for policymakers and pharmaceutical companies. CEA offers a range of benefits for the treatment of SMA, including improved resource allocation, informed decision-making, sustainability of healthcare systems, and promoting equity in access to care. By systematically evaluating the costs and benefits of SMA therapies, CEA helps ensure that resources are used efficiently and that patients receive the most effective treatments at a sustainable cost. As the availability of SMA therapies continues to expand, conducting CEA will be crucial in shaping policies that balance innovation with the practical realities of healthcare funding. In order to measure health-related quality of life to support economic evaluations in SMA population, Richard. H. Xu et. al. conducted measurement properties of the EQ-5D-5L and Patient-Reported Outcomes Measure Information System Preference measure(PROPr) in patients with SMA[33]. Strong ceiling and floor effects were observed for four dimensions of the EQ-5D-5L and three subscales, including pain intensity, pain interference, and physical function, of the PROMIS-29. The same group also conducted the measurement properties of the EQ-5D-Y-3L, PedsQL 4.0, and PROMIS-25 Profile v2.0 in pediatric patients with SMA[34].
We also cited the relative references in our manuscript.
- I recommend the authors revise the discussion thoroughly and maintain a more modest tone given the small sample size of 37, which may introduce significant bias. This study appears to be a pilot, and such strong statements should be reserved for more comprehensive formal studies.
Responses: Thanks for your comments. We really appreciate for your comments. The sample size is 37 and we agree that the size is still too low. According to your advice, we have checked again the discussion section and we maintain a more modest tone.We show some of the changes below:
For example, we have changed “confining most of the people at home” into “hard to visit the hospitals”.
We have removed words like “extremly”.
“After risdiplam was included in the reimbursement list in China, patients began to favor its use over nusinersen, primarily due to its ease of administration.”into “After risdiplam was included in the reimbursement list in China, it seems that patients began to favor its use over nusinersen, primarily due to its ease of administration.”
“In contrast, commercial insurance policies in developed countries often limit reimbursement for the combination of drugs for rare diseases [26].”into “In contrast, commercial insurance policies in some developed countries often limit reimbursement for the combination of drugs for rare diseases [26].”
“In contrast, caregivers in other countries typically remain employed, though they may experience absenteeism or reduced work hours [32].” into “In contrast, caregivers in some countries typically remain employed, though they may experience absenteeism or reduced work hours [32].”

Reviewer 4 Report
Comments and Suggestions for Authors
Dear Authors,
I found your article on healthcare resource utilization, economic burden, and the multi-level medical security system for individuals with Spinal Muscular Atrophy in Shaanxi province, China to be very interesting. However, there are several major issues that need to be addressed:
- Line 53: The abbreviation "NRPL" is not explained. It would be beneficial to clarify this term for the readers.
- There should be more detailed information on the current reimbursement systems. Specifically:
- Are the HTA criteria the same for orphan and non-orphan drugs?
- Is budget impact considered in these assessments?
- Are patients and advocates included in the HTA process?
- What are the cost-effectiveness thresholds, and is there a difference in these thresholds for rare diseases?
- Which of the three pillars (e.g., government, insurance, others) pays for the medicine?
- Is there a different reimbursement scheme, or if a drug is reimbursed, is it funded regardless of real-world evidence?
- Are there regional differences in the reimbursement systems?
- Is one of these three pillars mandatory, and if so, is it funded by general tax or mandatory health insurance contributions?
- Supplementary Table 2: The two tables should be separated, and there is no currency unit indicated for Supplementary Table 2. This should be clarified.
- Line 84: The abbreviation "MASC" is not explained. Please provide the full form or explanation.
- Information about ethics is mentioned in the supplementary file, but no such file was provided. When was this ethical decision obtained?
- Methodology Clarity:
- Initially, an online survey is mentioned (line 78), but then the data collection process is described with two investigators facilitating responses (line 102), which suggests a structured interview.
- Later, it was stated that the survey was available online again, and the parents were entering the data. The methodology should be more clearly outlined.
- Survey Questionnaire:
- The questionnaire used should be clearly described. Are there any translated versions in English?
- Why were validated item costs not used?
- Was there a list of prices for all cost items used to calculate direct and indirect costs? How was this list compiled, and were there reference prices?
- Did the price differ for individuals with "commercial" insurance?
- Insurance:
- The difference between commercial insurance and inclusive commercial insurance is not clearly explained.
- The cost of this insurance scheme, and whether it was paid by the parents, the state, or the employer for the whole family, should be clarified.
- Caregiver Productivity Loss:
- Was productivity loss or opportunity cost considered for caregivers of retirement-age individuals?
- Disability Grades:
- The tables mention disability grades (1, 2, etc.). What are the differences in terms of social support (money, insurance, or assistance paid by the state) in these grades?
- Was this support considered when estimating the economic burden?
- Civil Relief:
- The authors mention civil relief. Was this related to financial or other support from the government, and to what extent was it included in the economic burden estimate?
- Figures 1 and 2:
- There is a discrepancy between Figure 1 and Figure 2. Figure 1 indicates that the costs were selected only for 2023, but Figure 2 presents data for 2021-2023. Please clarify this inconsistency.
- Limitations:
- The paper does not have a designated limitations section. There are several points in the study that could be acknowledged as limitations by the authors.
- Prevalence and Sample Representation:
- Do you have information on the prevalence rate of Spinal Muscular Atrophy? Is your sample representative of the population, even partially?
- Second Table in Supplementary File:
- The second table in the supplementary file was not cited in the main body of the paper. It would be helpful to reference it appropriately.
- Opportunity Cost Adjustment:
- In line 119, was the opportunity cost adjusted for the educational level of the mother, or were reference hourly wages used? If reference wages were used, what was the value of those hourly wages for each year of the study?
- Statistical Reporting:
- I suggest presenting all results as medians with min-max ranges rather than means and standard deviations, not only to avoid skewness but also due to the small sample size.
- Line 130: The authors state that for skewed distribution, SD will be used with the median. This seems like a potential mistake. Please clarify this approach.
Author Response
REVIEWER4:
I found your article on healthcare resource utilization, economic burden, and the multi-level medical security system for individuals with Spinal Muscular Atrophy in Shaanxi province, China to be very interesting. However, there are several major issues that need to be addressed:
1.Line 53: The abbreviation "NRDL" is not explained. It would be beneficial to clarify this term for the readers.
Responses: We are sorry for the mis-understanding. We have explained the abbreviation in the abstract section. NRDL is short for National Reimbursement Drug List. We have checked again all other similar problems.
2.There should be more detailed information on the current reimbursement systems. Specifically:
Responses: We really appreciate for your comments. According to your comments, we have added more about the reimbursement systems in the introduction section. Please find the words in the red color in the second paragraph of the introduction section. We reply each of the question below:
- Are the HTA criteria the same for orphan and non-orphan drugs?
Responses: When HTA evaluates orphan drugs, the assessment process differs from that of non-orphan drugs due to their unique characteristics. The evaluation considers aspects such as clinical effectiveness and social value. For example, in the case of drugs treating SMA, clinical trials of nusinersen and risdiplam have shown that they significantly improve patients' motor function and slow disease progression. Since SMA patients are mostly children, treatment has a profound impact on their future lives, giving these drugs high social value. Real-world data further validate the long-term efficacy and safety of nusinersen and risdiplam, strengthening the evidence for its inclusion in healthcare reimbursement systems.
- Is budget impact considered in these assessments?
Responses: Through national drug price negotiations(In 2018, China introduced a policy called Volume-based Procurement (VBP), which significantly impacted the price negotiation process. Under VBP, the government conducts bulk procurement of drugs through a competitive bidding process. The goal is to reduce drug prices through competition.), nusinersen and risdiplam were included in China’s NRDL in 2022 and 2023, respectively [14,15]. Indeed, the Chinese government has a budget in place when conducting drug price negotiations.
- Are patients and advocates included in the HTA process?
Responses: Yes, the patients and advocates are included in the HTA process.
- What are the cost-effectiveness thresholds, and is there a difference in these thresholds for rare diseases?
Responses: The cost-effectiveness threshold for SMA treating medicines is higher than other medicines based on the current policies.
- Which of the three pillars (e.g., government, insurance, others) pays for the medicine?
Responses: After being included in the health insurance system, most patients are now able to receive treatment. The payment for medication costs includes the government, commercial insurance, social assistance, and individuals. The commercial insurance and social assistance have limited role in covering medication costs.
- Is there a different reimbursement scheme, or if a drug is reimbursed, is it funded regardless of real-world evidence?
In China, real-world evidence is not necessarily required for a drug to be included in the NRDL. HTA evaluations based on RCT (Randomized Clinical Trial) materials are also a method used to determine whether a drug can be included in the NRDL.
- Are there regional differences in the reimbursement systems?
Responses:For the treatment of SMA, the income and expenditure levels of medical insurance vary across different regions, leading to differences in reimbursement rates.
- Is one of these three pillars mandatory, and if so, is it funded by general tax or mandatory health insurance contributions?
Responses: The main sources of China's medical insurance funds are individual contributions from employees and urban and rural residents, employer contributions, and government fiscal subsidies. These funds collectively form the foundation of the medical insurance system, which is used to meet the healthcare needs of insured individuals.
3.Supplementary Table 2: The two tables should be separated, and there is no currency unit indicated for Supplementary Table 2. This should be clarified.
Responses: We really appreciate for your comments. We have added the currency unit in the table and the two tables have been separated.
4.Line 84: The abbreviation "MASC" is not explained. Please provide the full form or explanation.
Responses: We really appreciate for your comments. We have explained MASC in the manuscript. Meier Advocacy and Support Center is short for MASC.
5.Information about ethics is mentioned in the supplementary file, but no such file was provided. When was this ethical decision obtained?
Responses: We really appreciate for your comments. We have submitted the ethics to the editors and the final version of the submission will show the ethical decision.
6.Methodology Clarity:
- Initially, an online survey is mentioned (line 78), but then the data collection process is described with two investigators facilitating responses (line 102), which suggests a structured interview.
- Later, it was stated that the survey was available online again, and the parents were entering the data. The methodology should be more clearly outlined.
Responses: We really appreciate for your comments. We are sorry for the naïve mistake. Actually, these data are related to resource utilization for SMA patients between 2021 and 2023. The survey was conducted in October 2024. We have corrected it in the data collection section.
7.Survey Questionnaire:
- The questionnaire used should be clearly described. Are there any translated versions in English?
Responses: We really appreciate for your comments. We believe that including the questionnaire in the article is somewhat redundant and unnecessary to include as an appendix. Additionally, since healthcare reimbursement policies vary significantly across countries, it lacks broader relevance for promotion.
- Why were validated item costs not used?
Responses: We really appreciate for your comments. We don’t quite understand your question.
- Was there a list of prices for all cost items used to calculate direct and indirect costs? How was this list compiled, and were there reference prices?
Responses: We really appreciate for your comments. We have added more details about how to calculate the prices in the Statistical Analysis section. We show it again here:
We inquired about the annual direct costs for each patient, as well as their utilization of healthcare resources, including the number of hospitalizations, outpatient visits, and medication administrations. We calculated the total direct costs by multiplying the frequency of healthcare resource utilization by unit prices. In China, drug unit prices are publicly available, and the cost of a single hospitalization for SMA medication is approximately 5,000 CNY (700 USD). Outpatient visits primarily involve registration fees for prescription refills, with minimal costs that can be neglected. The costs of hospitalizations due to complications were obtained by directly asking the patients.
- Did the price differ for individuals with "commercial" insurance?
Responses: The price is the same for individuals with "commercial" insurance.
8.Insurance:
- The difference between commercial insurance and inclusive commercial insurance is not clearly explained.
Responses: We really appreciate for your comments. Commercial insurance typically excludes coverage for certain diseases, such as SMA, from its protection scope. In contrast, inclusive commercial insurance generally imposes no disease restrictions, features low premiums, and has low entry barriers. Although it offers broad coverage, the reimbursement rate is usually relatively low (e.g., 70%-80%), and it often includes a high deductible (e.g., 10,000–20,000 CNY or 1,400–2,800 USD).
We have included the above description in the manuscript at the beginning of the results section.
- The cost of this insurance scheme, and whether it was paid by the parents, the state, or the employer for the whole family, should be clarified.
Responses: Typically, the commercial insurance doesn’t cover disease such as SMA. The premiums for basic medical insurance and inclusive medical insurance are relatively low. The cost of the insurance scheme was paid by the patients’ parents.
9.Caregiver Productivity Loss:
- Was productivity loss or opportunity cost considered for caregivers of retirement-age individuals?
Responses: We really appreciate for your comments. The primary caregivers of SMA patients are typically their parents, with mothers being the main caregivers in most cases. Grandparents may also assist in caregiving, but they are not the primary caregivers. Since SMA patients are generally young, their parents are usually not yet of retirement age.
The cost of individual productivity loss is calculated based on the per capita annual income in Shaanxi Province.
We have added this in the manuscript.
10.Disability Grades:
- The tables mention disability grades (1, 2, etc.). What are the differences in terms of social support (money, insurance, or assistance paid by the state) in these grades?
- Was this support considered when estimating the economic burden?
Responses: We really appreciate for your comments. In China, the subsidies or assistance available to people with disabilities vary across different regions. Typically, people with disabilities can receive living subsidies, nursing care subsidies, medical assistance, and minimum living guarantee or special hardship assistance. In Shaanxi Province, the living subsidies and nursing care subsidies are very minimal and can be considered negligible. In 2023, the rural minimum living guarantee standard in Shaanxi Province is approximately 6,000 CNY(840USD) per person per year, while the urban minimum living guarantee standard is about 8,000 CNY(1120USD) per person per year.
Thus, we can see that this part of support is very little compared to the economic burden. We also added this into the manuscript.
11.Civil Relief:
- The authors mention civil relief. Was this related to financial or other support from the government, and to what extent was it included in the economic burden estimate?
Responses: We really appreciate for your comments. We have considered civil relief in our study.
Civil Relief includes policies for the rehabilitation of children with disabilities and public welfare funds such as the JD.com Rare Disease Challenge Foundation. Among these, the rehabilitation policy for children with disabilities in Xi'an has a low entry threshold, targeting children aged 0–16 with conditions such as cerebral palsy, intellectual disabilities, autism spectrum disorder, and SMA, providing 28,000 CNY(3,920USD) per child per year. The JD.com Rare Disease Challenge Foundation, which serves patients nationwide, offers an average assistance amount of approximately 20,000 CNY(2,800USD) per case, and applications can be submitted annually. The funds are primarily used to cover the costs of specialized medications for rare diseases like SMA. This fund also has a low entry threshold, with a focus on assisting urban and rural recipients of the minimum living guarantee. The civil relief was also included in the economic burden estimation. In the questionnaires, social relief was included.
We have added the above talk in our manuscript. Please find them in the results section.
12.Figures 1 and 2:
- There is a discrepancy between Figure 1 and Figure 2. Figure 1 indicates that the costs were selected only for 2023, but Figure 2 presents data for 2021-2023. Please clarify this inconsistency.
Responses: We really appreciate for your comments. The reason why we only include 2023 in figure 1 is because the availability of both nusinersen and risdiplam during that year.2023 could represents the real status of resource usage of the patients. We have described the reasons in the manuscript. In figure.2, we have list all the cost from 2021 to 2023. We want to show more details of the cost for different grades and years. This could also shows the change of cost with the including of medicines in the NRDL.
13.Limitations:
- The paper does not have a designated limitations section. There are several points in the study that could be acknowledged as limitations by the authors.
Responses: We really appreciate for your comments. We have a limitation description in the manuscript. We didn’t list it as a separation section. We list it at the end of the discussion. We show it again here:
A key limitation of this study is the retrospective nature of the data collection, which introduces the potential for recall bias. Nevertheless, we examined the utilization of both nusinersen and risdiplam. Since nusinersen is administered via intrathecal injection and has been in use for only two years (with a standard dosage of six injections in the first year, followed by a maintenance dose of four times per year), the recollections of patients' parents are likely to be relatively accurate. Additionally, we incorporated data from the Shaanxi Provincial Healthcare Security Bureau to conduct relevant analyses on nusinersen use, which enhances the robustness of our survey results (see Supplementary Table 2).
14.Prevalence and Sample Representation:
- Do you have information on the prevalence rate of Spinal Muscular Atrophy? Is your sample representative of the population, even partially?
Responses: We really appreciate for your comments. We currently do not have data on the incidence rate of SMA disease in Shaanxi Province. We believe that our sample could represent partially the population.
15.Second Table in Supplementary File:
- The second table in the supplementary file was not cited in the main body of the paper. It would be helpful to reference it appropriately.
Responses: We really appreciate for your comments. We have cited the supplementary table in the main manuscript. Please see the end of the last paragraph at the discussion section. We show it again here.
“Additionally, we incorporated data from the Shaanxi Provincial Healthcare Security Bureau to conduct relevant analyses on nusinersen use, which enhances the robustness of our survey results (see Supplementary Table 2).”
16.Opportunity Cost Adjustment:
- In line 119, was the opportunity cost adjusted for the educational level of the mother, or were reference hourly wages used? If reference wages were used, what was the value of those hourly wages for each year of the study?
Responses: We really appreciate for your comments. The opportunity cost is not adjusted according to the caregiver’s education level. We used the reference wages. The cost of individual productivity loss is calculated based on the per capita annual income in Shaanxi Province. We have added the calculation method in the manuscript.
17.Statistical Reporting:
- I suggest presenting all results as medians with min-max ranges rather than means and standard deviations, not only to avoid skewness but also due to the small sample size.
Responses: We really appreciate for your comments. Thanks for your advices. We have considered this problem during the design of the study. We have shown both the means and median value of the statistical reporting. Please see the tables in the manuscript. Take table.4 for example, both mean and median values are listed.
18.Line 130: The authors state that for skewed distribution, SD will be used with the median. This seems like a potential mistake. Please clarify this approach.
Responses: We really appreciate for your comments. We agree that the SD will not be used with the median values. SDs will be used with the mean values. We have checked again the manuscript. As we use the mean values, the SDs were listed. We also gave the median values in the tables for clarities.

Round 2
Reviewer 3 Report
Comments and Suggestions for Authors
All my concerns have been addressed. Thanks